# Analysis of Adhesion at the Interface of Steamed Bread and Eggshell

**DOI:** 10.3390/molecules27238179

**Published:** 2022-11-24

**Authors:** Qunfeng Zeng, Jianing Zhu

**Affiliations:** Key Laboratory of Education Ministry for Modern Design and Rotor-Bearing System, Xi’an Jiaotong University, Xi’an 710049, China

**Keywords:** adhesion, starch, diffusion, multiple hydrogen bonds, molecular dynamic

## Abstract

The adhesion phenomenon of polymers occurs in nature and in human activity. In the present paper, an adhesion system of steamed bread and eggshell was observed in formation when steamed bread and eggshells were placed in close contact and cooled slightly in the ambient air. The adhesion phenomena and mechanism of the adhesion interface between the steamed bread and eggshell were investigated and systematically discussed. Strong-bond interfaces were observed by scanning electron microscope (SEM). The formation process and mechanism of the strong-bond adhesion were also analyzed molecular dynamics simulation technology, and the results are discussed. The simulation analyses showed that the starch molecules at the calcite (104) crystal face were diffused in a water vapor environment, and the formation and solidification of multiple hydrogen bonds in the starch chain and oxygen atoms in the calcium carbonate were observed in detail during cooling. The diffusion rate of hydrogen atoms in hydroxyl groups on the calcite surface decreased gradually with the decrease of the cooling temperature of the steamed bread’s upper surface. The strong adhesion of the steamed bread and eggshell is attributed to the synthetic effect of the absorption, diffusion, surface chemistry, and the formation of multiple hydrogen bonds between the starch from the steamed bread and the calcium carbonate crystals in eggshell. The interesting findings are helpful for the design of strong bonds, and provide an idea for new environmentally friendly adhesive materials.

## 1. Introduction

Adhesive bonding technology has become popular in recent years due to its ability to bond various types of materials [1]. It is a good alternative method to conventional techniques that including mechanical fastening, welding, and soldering. Adhesive bonding is applied in many fields including automotive, aerospace, and electronic engineering [2]. Adhesive materials are often used to bond various substrates. The bond structures and adhesion mechanism are important for enhancing the mechanical properties of the film–substrate system [3,4]. The production processes of traditional adhesive materials used in construction or other engineering applications, such as cement orpolyvinyl alcohol (PVA) [5,6], produce many hazardous by-products including coal slag, dust, NO_x_ and other pollutants. Moreover, these traditional materials are difficult to degrade by pollution-free treatment after use [7,8]. It is of major interest to identify and develop degradable or nonhazardous materials for use as adhesive materials with good physical and chemical properties and favorable surface properties such as adsorption and adhesion [9,10]. Among the natural polymers, starch occurs as a natural product and renewable material, and has been of particular interest due to its positive tendencies concerning degradability and physiological characteristics. Owing to its complete biodegradability, low cost, and renewability, starch is considered as a promising candidate for developing sustainable materials. Starch is found primarily in the seeds, fruits, and stem pith of plants, most notably corn, wheat, rice, sago, and potatoes. Najafi et al. studied the fracture behavior between latex starch adhesive and calcium carbonate, and found that samples containing starch had good bonding properties [11]. Starch has been used as an adhesive in a wide range of products, including binders, sizing material, glues, and pastes [12]. Starch meets the wide range of the performance requirements and can provide high performance adhesives. In some cases, starch and starch derivatives have replaces synthetic adhesive raw materials. Starch-based adhesive is one of the formaldehyde-free adhesives. Clearly, this makes starch as a very important adhesive material and a useful binder of pigment such as calcium carbonate. Calcium carbonate, which is chemical compound with a chemical formula of CaCO_3_ that is extensively present in shells of microorganisms and in eggs, is one of the most common and profuse biominerals present across the world and is widely employed as a filling material in paper and adhesive industries [13]. Adhesion is one of the most important and difficult tasks of surface engineering. In spite of immense potential fundamental and industrial applications, one of challenges is how to improve the binder–pigment adhesion in terms of the crack resistance of the binder–pigment system. It is known that numerous parameters have an influence on adhesion, including chemical bonding and the physical properties of adhesive materials. However, there has been little research into the physical–chemical adhesion of the attractive binder–pigment system within an organic–inorganic system, to elucidate the theory of polymer adhesion from the physical and chemical standpoints.

In the present research, to study the adhesive behavior of starch, bread was heated by high-temperature water steam and then cooled with heated egg in ambient air.The adhesion phenomenon between starch from the steamed bread and calcium carbonate from the eggshell was observed, and the bonding interface with strong adhesion may be beneficial for designing strong-bond interfaces and understanding the interface formation of organic and inorganic materials. This research can provide ideas for expanding the application prospects of starch in environmentally friendly coatings and bonding materials.

## 2. Results and Discussion

### 2.1. Characterization of Interface

Figure 1 shows the SEM images of the bonded sample of the steamed bread and eggshell.. The eggs were cooked in the water and the bread on the steamer, and the water heated to boiling for 15–20 min. After heating, the eggs and steamed bread were taken out, and then placed in close contact with each other to cool in the ambient air. After cooling, they stuck together. The two structures can be distinguished in the SEM images shown in Figure 1a. The 90-fold overall topography of the adhesion interface between the steamed bread and eggshell is shown in Figure 1a. It can be seen that the eggshell is a highly ordered mineralized structure consisting of calcite crystals. The steamed bread is made from the fermented dough and has a loose porous structure. The adhesion interface is shown in Figure 1b, which shows the electron micrograph of the adhesion between the steamed bread and eggshell at 7000-fold. The steamed bread was easily absorbed in the holes on the surface of the eggshell. Figure 1c shows a local high-magnification SEM image of the interface between the steamed bread and egg shell at45,000-fold. The skin of the steamed bread is firmly attached to the surface of the eggshell and the two cannot be separated.

### 2.2. Microstructure Analysis 

The microstructure of the adhesion interface of the steamed bread and eggshell was measured using FTIR, to study the interesting scientific phenomenon. As shown in Figure 2, a hydroxyl absorption peak is present at a wavenumber of 3274 cm^−1^ and the peak shape is broad [14,15]. A very sharp and intense absorption is centered at 2348 cm^−1^. The characteristic spectral signature of the antisymmetric stretching mode of C=O yields a fairly strong peak at 2348 cm^–1^. A medium-intensity peak at 2928 cm^−1^ is the antisymmetric stretching vibration of −CH_2_−. The absorption peak at 1640 cm^−1^ is the absorption peak of the amorphous region in the absorbed water in the starch [16]. The activity at 1529 cm^−1^ can be noted, which may originate corresponding to the wheat flour peak region of proteins [17]. The absorption peak at 1443 cm^−1^ is −CH_2_− and there is a C–H in-plane bending vibration at 1068 cm^−1^ with strong absorption intensities. The FTIR spectroscopy of the test sample based on absorbance at 1410 cm^−1^ (nCOO−) was associated with the aromatic ring vibration. A band at 1239 cm^−1^ was observed, relating to the stretching acyl groups in egg albumen [18]. Therefore, it can be inferred that there are many hydrogen bonds in hydroxyl groups and other groups such as carbonyl groups and hydrocarbon at the interface of the steamed bread and eggshell. It was at this stage necessary to investigate the formation of hydrogen bonds and the influence of hydroxyl groups on the adhesion mechanism of the steamed bread and eggshell.

### 2.3. Analysis of AdhesionInterface

Adhesion at the interface between the steamed bread and eggshell exists as shown in Figure 1 and Figure 2. The factors of adhesion occurring during the cooling process of the steamed bread and eggshell are governed by several possible mechanisms, including adsorption, diffusion, and surface chemistry, which must be understood to interpret the phenomenon of interest.

Firstly, adsorption was critical to the adhesive interaction because the water vapor on the steamed bread skin was in contact with the eggshell under high temperature and pressure when the steamed bread and eggshell were close. Starch is a natural hydrophilic polymer. The flour in the bread can easily absorb up to more 50% water molecules on a dry weight basis when steamed in water vapor. Steamed bread was used in which the starch present in the flour becomes sticky during the manufacturing process. During the experiment, it was treated a second time with steam. During steaming of the bread, the top surface of the starch was wet with water vapor. It would seem that water vapor from the upper surface of the steamed bread and the surrounding air entered the surface of the eggshell by seepage at the interface after the surfaces made contact, and were possibly easily condensed in the eggshell due to the contact pressure. To form an adhesive bond, the first step requires interfacial molecular contact due to wetting, then the molecules diffuse across the interface or even react chemically to establish covalent bonds. Thus, water penetration through interfaces is associated with heat transport and consequently water diffusion into the adhesive through the interface. The steamed bread sticks to the eggshell by absorption and diffusion with the epidermal moisture. 

Theoretically, the absorption processes is described by Fick’s second law of diffusion [19,20]. The initial stage of absorption (*M*_t_/*M*_∞_ < 0.5) is as follows:(1)$$\frac{M_{t}}{M_{\infty }}=4{\left(\frac{Dt}{\pi h^{2}}\right)}^{1/2}\;$$
where *M*_t_ and *M*_∞_ are the moisture contents at time *t* and at equilibrium, respectively. *D* is the diffusion coefficient and *h* is the thickness. The diffusion coefficient of water in starch reduces because the water content evaporates gradually into the air. With increase of the cooling time, the temperature and water vapor content of the steamed bread decrease, thus the absorption distinctly decreases. Note that the moisture absorption increases linearly with *t*^1/2^ in the initial stage, then the increase rate slows down and finally leads to equilibrium. Fick’s second law, which predicts change in concentration with time due to diffusion, is a parabolic partial differential equation. The diffusion coefficient *D* is an important physical quantity describing the diffusion velocity. The larger the value of *D*, the faster is the diffusion. This equation describes the change of the concentration of matter at each point in the medium due to diffusion, under conditions of unstable diffusion. The law of material concentration variation with time and location can be obtained by solving Fick’s second diffusion equation according to the initial conditions and boundary conditions. The data of time and concentration were applied to Equation (1). If the absorption process occurs in the atmosphere at moderate relative humidity, the equilibration time may be even longer. This analysis indicates that the absorption of water by the starch may significantly hinder diffusion.

In this contact system, surface chemistry may give rise to adhesion under high temperature and pressure. There are three steps to making steamed bread: kneading the flour into the dough with the yeast water; fermentation; and steaming the bread. An certain amount of acids including lactic acid may be generated on the surface of steamed bread during fermentation and steaming, affecting the pH value of the steamed bread’s skin. The lactic acid probably resulted in primary chemical bonds with calcium carbonate in the present contact system, leading to high interface adhesion, as shown in Equation (2).
2CH_3_CH(OH)COOH + CaCO_3_ = Ca(CH_3_CH(OH)COO)_2_ + CO_2_↑ + H_2_O (2)

The diffusion increases the adhesive bond strength. As discussed above, diffusion and absorption are two critical factors of adhesion. Diffusion can be divided into two parts: water absorbed on the bread skin diffuses into the surface of the eggshell, or seeping water spreads close to the interface at a slow rate. Absorption was determined by water diffusion at the adhesion interface. The diffusion was related to the presence of strong hydrogen bonding interactions between starch and eggshell at the interface. Therefore, molecular dynamics simulations were employed to analyze the formation of hydrogen bonds in different hydroxyl groups. This simulation provided information about diffusion at the interface, allowing molecular insight into the adhesion mechanism.

### 2.4. Molecular Dynamics Simulations of the Formation of Hydrogen Bond 

According to diffusion theory, it is difficult to obtain high adhesion strength when the interfacial molecules are in directly contact with each other, and it is necessary to create interfacial diffusion between the polymer molecular chains. The diffusion involves a network formed by adhesion at the interface, which leads to the disappearance of the interface and the generation of a transition zone, and the two phases are connected through the winding of the diffused molecules or chain segments and their cohesion to form a strong adhesive joint. Macroscopic physical quantities are predicted and calculated by the molecular dynamics method through simulating the microstructure and motion of the system’s molecules [21,22,23]. An image of the microstructure, particle motion and the relationship with the macroscopic properties of material can be provided, and it can reveal the macroscopic properties and processes of many substances. Molecular dynamics simulation experiments were conducted using the For cite module of Materials Studio software, to study the bonding mechanism between the starch and calcium carbonate [24].

The molecular dynamics simulations were employed to analyze the formation of hydrogen bonds in the steamed bread and eggshell. The simulation provided information about the diffusion at the interface, allowing for molecular insight into the adhesion mechanism. In order to study the diffusion rate of the starch molecules on the surface of calcium carbonate under different temperatures, the mean square displacement of hydrogen atoms in starch molecules was calculated to obtain the coefficient of diffusion and characterize the compatibility between starch and calcium carbonate. Therefore, the mean square displacement (MSD) curves of hydrogen atoms in starch molecules at these temperatures were obtained from the molecular dynamics simulations.

The main ingredient of the steamed bread is starch [25], which is a high-molecular polymer composed of monosaccharides (glucose) [26] and contains a large number of dangling hydroxyl groups in the molecular chain (Figure 3a). Eggshell has a highly mineralized structure composed of calcite crystals with many stomata distributed on the surface (Figure 3b). According to the measurements taken by energy dispersive spectroscopy (EDS), the elements contained in the eggshell were mainly calcium, carbon, and oxygen (mainly CaCO_3_) (Figure 3c) [27]. Optimized five-layer calcium carbonate has a molecular size of 24.26 Å × 24.95 Å, (104) and the crystal face is used as the adsorption plane of the starch molecule [28,29,30].

A supercell was created by replicating the (104) crystal face three to five times the size of the unit cell in the X and Y directions. The 3D crystal slab was created by five layers of calcium carbonate molecules (Figure 3d). A fixed constraint was applied to the calcite crystal. The structure of starch, shown in Figure 4a, consists of five glucose molecules. The mechanism of the functional groups in starch was simulated using a polymer model formed by the dehydration condensation reaction of five glucose molecules. This structure was optimized using the smart algorithm and the COMPASS force field. The initial distance between the starch molecule and the calcite crystals was4.0 Å. A vacuum space on top of the starch molecule of 40 Å was established to restrict its interaction with the uppermost atom layer of the starch and to enhance computational efficiency [31].

Ten water molecules ware added in the model to simulate the water environment in the steamed bread. The molecular dynamics model is shown in Figure 3d. The COMPASS force field was employed to simulate the interaction between polymers and inorganic molecules. The COMPASS force field can accurately and reasonably represent hydrogen bonding [32]. In the present work, the critical distance for hydrogen bonding was set to 2.5 Å [33].

Calcium carbonate crystals were placed on the bottom and fixed at the beginning of the simulation. Periodic boundary conditions were applied in the plane containing the interface of the two layers, to simulate an infinite contact area. The whole model was simulated under the NVT ensemble and the simulation temperatures were set to 323.15 K, 333.15 K, 343.15 K, 353.15 K, 363.15 K, 373.15 K, and 393.15 K in order to simulate the adhesion between the steamed bread and egg shell. In Figure 4c, the axis represents the acceleration voltage of the excited element. Figure 4c shows that the eggshell had a highly mineralized structure composed of the calcite crystals with many stomata distributed on the surface. According to the measurements of EDS, there were elements of carbon, calcium, and oxygen at the adhesion interface.

The diffusion was analyzed theoretically to understand the structure of the solids, the state of bonding of atoms, and other mechanisms. In order to study the diffusion rate of the starch molecules on the surface of calcium carbonate under different temperatures, the mean square displacement (MSD) of hydrogen atoms in starch molecules was calculated to obtain the coefficient of diffusion and characterize the compatibility between starch and calcium carbonate [34]. Therefore, the mean square displacement (MSD) curves of hydrogen atoms in starch molecules at these temperatures were obtained (Figure 4a) from the molecular dynamics simulations. It is concluded from the curves that in the first 30 ps, there was no equilibrium state during the diffusion process and the values were unstable. After 30 ps, the MSD curve is a straight line; the curve after 30 ps is linearly fitted and the slope of the straight line was calculated (Figure 4b).

According to the formula of the diffusion coefficient and MSD (Nandi et al., 2012) [35]:(3)$$D=\underset{t\to \infty }{\lim}\frac{1}{6t}\langle {\left|r(t)-r(0)\right|}^{2}\rangle$$
where *D* is diffusion coefficient; *t* is time; |*r*(0) − *r*(*t*)|^2^ are the square values of atomic displacement. The diffusion coefficient is one sixth of the slope in Figure 4b, and is shown in Table 1. The diffusion coefficient depends on many factors; diffusion is affected by the contact time and bonding temperature. When polymers are bonded to each other, higher temperatures and longer time generally lead to stronger diffusion. The molecular diffusion coefficient represents the diffusion capacity. According to Fick’s law, the diffusion coefficient depends on the type of the diffused substance, the diffused medium, and its temperature and pressure.

The interaction energy of the steamed bread and eggshell was calculated to determine the stability of the bonding interface at different temperatures. The intermolecular interaction energy was calculated as the total energy of each system under a stable configuration. The interaction energy was calculated using the following Equation (4).
*E*_interaction_=*E*_total_ − (*E*_surface_ + *E*_polymer_)(4)
where *E*_total_ is the energy of the surface of the calcium carbonate and the starch molecules; *E*_surface_ is the energy of the surface without the starch molecules; *E*_polymer_ is the energy of the starch molecules without the surface. Surface energy is an important parameter, which is required to break the chemical bonds between molecules when creating a material surface. Surface energy is an important factor affecting adhesion strength. More heat is required to cut through surface atoms, as surface atoms have higher heat than atoms inside a substance. The surface-energy calculation method requires the structural optimization of the slab model after cutting. The principle of surface-energy calculation is that the energy required to cut the block into crystalline surfaces is equivalent to the energy required to form two new surfaces. In the calculation of surface energy, the upper and lower surfaces should be optimized at the same time. The intermolecular interaction energy was calculated as the total energy of each system under a stable configuration. The action of bonding occurs at interfaces. Good wetting and bonding are necessary to achieve the formation of good adhesive force. Reducing the surface energy or increasing the surface energy of the adhesive can enhance the wettability of the adhesive, thus improving the adhesive strength. Many environmental conditions affect the performance of adhesives. One of the most important environmental factors is temperature. A small change in temperature can have a significant impact on the rate of adhesion. Temperature affects the bonding strength of adhesives; the higher the temperature is, the faster the glue dries.

Figure 5 shows the interaction energy at different temperatures.

Dried starch is generally not sticky, because the molecules in the dried starch granules form an extremely strong associative state due to the intermolecular hydrogen bond [36]. As shown in Figure 6, the starch slurry in the steamed bread was heated and the water molecules entered into the amorphous regions of the starch granules, and then the hydrogen bonds were broken among the starch molecules to eliminate the association in the molecular chain. The intermolecular hydrogen bonds in the crystalline regions of the starch granules were destroyed under high temperature and the starch irreversibly and rapidly generated a viscous starch paste.

It was found that the diffusion coefficient *D* fluctuated from the simulation results slightly at 50 °C, as shown in Table 1. The diffusion coefficient increased suddenly at the temperature increase from 60 °C to 70 °C. At 100°C, the diffusion coefficient reached the maximum value of 1.58. However, the diffusion coefficient of the hydrogen atoms decreased at a temperature of 120 °C. The gelatinization of wheat starch is divided into three stages: (1) In the reversible water-absorption stage at relatively low temperatures, the water molecules are introduced into the amorphous part of the starch granules, and the hydrogen bonds in the crystal part are not destroyed. Therefore, this stage is reversible, and the starch particles can be recovered in the birefringence state after drying. (2) Water molecules are introduced into the microcrystalline area of the starch, and the starch irreversibly absorbs a large amount of water with the increase of temperature. The volume rapidly expands and the birefringence phenomenon gradually disappears. For wheat starch, birefringence disappears completely when it is heated to 65 °C. (3) The starch is completely dissolved in the solution. The increase in the diffusion coefficient at temperatures of 60–70 °C is related to the rapid increase of gelatinization of the wheat starch in the temperature range 60–70 °C. The diffusion coefficient reached its maximum value at 100 °C, at which point the diffusion coefficient and the diffusion rate of hydrogen atoms were at their highest. The compatibility between the starch and the calcium carbon at e surface was the strongest at this temperature, confirming a solid bonding between the eggshell and the steamed bread skin at a temperature of 100°C. However, at 120°C, the compatibility of the starch and calcium carbonate molecules decreased, which may be related to hydrogen bond cleavage at 120 °C.

The interaction energy is shown in Figure 6. The interaction energy gradually decreased with the increase of temperature, which means that the adhesion between the starch molecules and the calcium carbonate crystals occurred easily at high temperature and solidified stably at relatively low temperatures. The simulation results showed that adhesion between the steamed bread skin and eggshell occurred under high temperatures and solidified during cooling.

There are two types of hydroxyl groups in starch. One is a primary hydroxyl group attached to a primary carbon atom, and the other is a secondary hydroxyl group attached to a secondary carbon atom. The interaction energies of the two hydroxyl groups and the surface of the calcium carbonate crystals were calculated by the molecular dynamics method. According to Formula 2, the *E*_interaction_ between the surface of the calcium carbonate and the primary hydroxyl groups was −0.74 kcal·mol^−1^ and the *E*_interaction_ between the surface of the calcium carbonate and the secondary hydroxyl groups was −0.12 kcal·mol^−1^. The primary hydroxyl groups have higher interaction energy than the secondary hydroxyl groups. Therefore, the primary hydroxyl group bonded preferentially to form a stable hydrogen bond with the oxygen atom in calcium carbonate [37].

According to the 3D model, it can be seen that the hydroxyl groups in the starch along with the oxygen atoms in the calcium carbonate molecule formed into the hydrogen bond structures, and the schematic representation of the simulation process is shown in Figure 7. The hydrogen atoms and oxygen atoms in the hydroxyl groups of the D-glucose monomer are covalently bonded. The ability to attract valence electrons is strong, as oxygen atoms have large electronegativity. Therefore, the oxygen atoms in the hydroxyl group adsorb electrons in the hydrogen atoms to one side, making the hydrogen atoms almost bare-pore protons. Because the radius of the proton is extremely small, it can interact with the oxygen atoms in the calcium carbonate with a lone electron pair and large electronegativity to form a hydrogen bond. According to the simulation results, only hydrogen atoms in the primary hydroxyl group reacted chemically to form hydrogen bonds with the calcium carbonate crystals before 18 ps (Figure 7a). At 18–30 ps, the hydrogen atoms in the secondary hydroxyl group were also preliminarily involved in forming hydrogen bonds. After 30 ps, according to the MSD curve, the system balanced towards equilibrium and the hydrogen bonds were formed by both types of hydroxyl groups (Figure 7b). The OH functional groups improved adhesion-forming organic–inorganic hybrid chemical bonds. A strong adhesion formed between the starch molecules and calcium carbonate crystals during cooling.

Adhesion is considered a reflection of the force necessary to rupture interface bonds when bodies are separated. The nature of adhesion has been studied extensively. There are many factors affecting the adhesion of eggshell and steamed bread, such as the components and structures of eggshell and bread, the water, the temperature, time, and environmental conditions, adsorption, diffusion, cold shrinkage in water, starch, and calcium carbonate, adhesion of proteins and glucose, and so on. As we know, eggshell is highly porous, as shown in Figure 1, a feature of the metabolism of life and of evolution. This porous structure is helpful for the adsorption and diffusion of water and starch into the eggshell. There are also many pores in bread, and water is easily diffused in steamed bread. The adhesion of a surface is strongly related to its wetting. The diffusion coefficient of water is high, and the moisture absorption is determined by the diffusion of water at the interface. Continued diffusion leads to the formation of a chemical bond between calcium carbonate atoms and water vapor atoms at the interface. In other words, there is a “bridge” of bonded atoms that connects the eggshell and skin of bread. Thus, the interface has achieved a steady-state structure and strong adhesive bonds can be formed between atoms on the contact surfaces. Adhesion depends somewhat on the nature of the polymer, but mostly on local physical conditions. Diffusion depends on many parameters (temperature, etc.); it has been shown that increasing temperature clearly increases diffusion. Temperature was found to bean important factor in the adhesion of eggshell and steamed bread. At high temperatures, the water vapor was diffused quickly, and the speed of diffusion and the water vapor content decreased with decrease of temperature. Where the eggshell and starch interact directly with each other, the starch molecules come into interplay with the others. In the case of long-chain polymer molecules, molecules of starch are adsorbed onto the eggshell surface. The adhesion is associated with physical and chemical contributions of the active chains of polymer molecules. Owing to the direct interaction with starch surfaces, physisorption and direct molecular bonding coexist at points of contact. The chemisorptions of polymer chains are explained by the appearance of strong chemical bonds at the contact points. 

Although hydrogen bonds are the dominant source of adhesion of the calcium carbonate and the starch, there may be other factors involved, such as the nonbonded van der Waals interactions, which are always attractive and are considered a source of adhesion. Adhesion may be also generated at the interface, by the capillary condensation of water vapor from the high relative humidity environment and from surface molecular irregularities interlocking at the interaction on the starch surface. If sufficient water is present to form a meniscus bridge, the number of contacting and near-contacting asperities forming meniscus bridges increases with an increase of humidity, leading to an increase in meniscus force. The combined effect of the contribution of van der Waals and meniscus forces increases the adhesive force. 

## 3. Experimental Materials and Methods

### 3.1. Materials and Experimental Details

There are many natural adhesive systems, such as gecko footpads (van der Waals forces), the secretions of mussels and tube worms (hydrogen bonds), and various biological processes including cell adhesion to extracellular bio-molecules, or bacterial adhesion [38]. This study paper reports research about the role of starch adhesives in nature as a source of inspiration. Steamed bread was reheated by high-temperature water steam from boiling water on a steam drawer, and the egg was cooked in the hot water in the steamer. Then, the steamed bread and the cooked egg we replaced together ensuring they were in direct as closely as possible, and finally they were cooled in the ambient air. An interesting phenomenon of strong bonds can thus be observed when breakfast food is cooked in everyday life. The experimental procedures are described here as follows. The steamed bread samples, which were bought from the local store, were reheated in a food steamer by high-temperature water steam. The steamed bread mainly contained carbohydrates, crude fiber, protein, and manganese as well as other substances. The eggs were bought at local supermarkets and generally consumed for breakfast. They were cooked in the boiling water in the steamer. After cooking, the steamed bread and cooked eggs were immediately taken out from the food steamer and placed together in contacted with each other as closely as possible, in a bowl in ambient air to cool for testing. The cooking temperature was around 100 °C, the time of exposure to steam was about 20 min, and the time for sufficient cooling was about 20 min in ambient air. The experiments were performed many times when the bread and eggshell were cooked every morning. It is interesting to observe that the strong bond between the top surface of the steamed bread cooked at high temperature and the shell of the cooked egg was firmly formed when the steamed bread and eggshell were in close contact and completely cooled to room temperature after 20 min in ambient air. In these experiments, the steamed bread and eggs were placed in a steam pot and heated. Then, the steamed bread and the cooked eggs were removed into the atmosphere and brought into contact with each other at room temperature. The edge of the bread was selected for sample preparation after cooling. Finally, the eggshell attached to the steamed bread’s upper surface was removed carefully to provide the test sample, as shown in Figure 8. Figure 8 shows the image of the steamed bread’s upper surface and the eggshell after cooling. It is well known that the steamed bread is made of wheat flour mixed with water and baking soda. After steaming, the steamed bread became loose and porous with plentiful water vapor on its skin. The two surfaces of steamed bread and eggshell stuck firmly and could not be peeled apart completely after cooling. A skin was formed on the steamed bread during steaming, and adhered to the surface of the eggshell during cooling when the steamed bread and eggshell were put in contact with each other. Starch is mainly composed of two homopolymers of D-glucose, i.e., amylase, a mostly linear glucan, and branched amylopectin, with the same backbone structure. There are many hydroxyl groups in the starch chain, among which are secondary hydroxyl groups connected to C-2 and C-3,as well as one primary hydroxyl group unlinked at C-6. In other words, they can be easily oxidized in ambient air especially at high temperatures, and may participate in the formation of hydrogen bonds, ethers, and esters. Starch granules exhibit hydrophilic properties and strong intermolecular association via hydrogen bonding formed by the hydroxyl groups on the granule surface [39]. Eggshell, a highly ordered porous ceramic, contains about 80% calcium carbonate, 15% proteins, and other microelements such as Zn, Cu, Mn, Fe and Se [40].

### 3.2. Analysis of Structure

After cooling, the morphology and structure of the sample were measured by scanning electron microscope (SEM, S-3000N, Tokyo, Japan) operating at 20 kV. The cross-section of the test sample was cooled to room temperature and then gold-sprayed. Qualitative observations of the microstructure and the interface between the steamed bread and the eggshell were made using SEM, to study its morphology in detail. The observations were made using a secondary electron detector. Firstly, the overall topography of the steamed bread and eggshell was measured by SEM to identify the adhesion interface at low resolution. Resolution is the most basic performance index of SEM. It was necessary that we should clarify some details using high resolution SEM, so the location of the adhesion interface obtained from low resolution topography was observed under high resolution at 7000-fold, which revealed that the steamed bread skin expanded into the holes on the surface of the eggshell and increased the contact area. Finally, the adhesion of the steamed bread and eggshell was observed under high resolution at 45,000-fold to obtain details of the adhesion interface. There are many types of vibrations in molecules, some of which can cause changes in molecular dipole distance. When the frequency of vibrations is the same as that of the infrared light, molecules can absorb the energy of the infrared light, forming the infrared absorption spectrum. Due to their different molecular structures, different compounds have different characteristic peaks of infrared absorption spectra. Like human fingerprints, no two compounds are completely consistent. Therefore, Fourier transform infrared spectrometry (FTIR) is considered a very effective method in the analysis and identification of polymer materials. This technique offers information about the vibration of chemical bonds that can be utilized to characterize the microstructure. In structural identification, quantitative analysis and chemical kinetics research, analysis can provide information about functional groups, the position and intensity of the infrared absorption peaks reflecting the characteristics of molecular structure, thus FTIR can be used to identify the structure of an unknown substance or determine its chemical groups. FTIR analysis is an important modern analytical method, and has been widely used to obtain material evidence in the judicial authentication of samples (including organic and inorganic materials) by qualitative and quantitative analysis. Not only it can accurately determine materials of all kinds of chemical composition, but it can also carry out comparative analysis quickly and efficiently. The microstructure of the adhesion interface of the steamed bread and eggshell was measured using FTIR, to study the interesting scientific phenomenon presented in this paper. The FTIR (Nicolet iS50, Thermo Fisher, Waltham, MA, USA) performed sample testing in the range of 800~4000 cm^−1^.

### 3.3. Molecular Dynamic Simulations

A molecular model of the starch–calcium carbonate system was established to study the bonding mechanism between starch and calcium, based on molecular dynamic methods. All molecular dynamic (MD) simulations were conducted with periodic boundary conditions and explicit treatment of all atoms in the structures. All charges applied throughout the whole system were based on the atom parameters from the COMPASS force field, and the entire model had 960 atoms. All MD simulations were conducted using the canonical NVT ensemble (constant number of atoms, volume, and temperature). The standard Verlet algorithm was applied to iterate the equations of motion in the system. The temperature was controlled using a Nosé–Hoover thermostat with a relaxation time of 0.1 ps. A 1 fs time step was used and all the simulations were run for a total of 100 ps.

## 4. Conclusions

The strong-bond case of the steamed bread’s upper surface and the eggshell was investigated systematically, and the adhesion mechanism based on the multiple hydrogen bonds in starch molecules has been scientifically discussed according to observations of the adhesion phenomena and molecular dynamics simulations. This mechanism can be successfully converted into a starch-based bonding material with potential value in engineering applications. The adhesion mechanism is attributed to the absorption, diffusion, surface chemistry, and the presence of the multiple hydrogen bond structure at the interface layer formed by hydrogen atoms in two different active hydroxyl groups, and oxygen atoms in calcium carbonate. The hydrogen atoms in the primary hydroxyl group in starch firstly form hydrogen bonds with the oxygen atoms in calcium carbonate at the initial interface, due to the strong activity of the primary hydroxyl, and then hydrogen atoms in the secondary hydroxyl group are bonded with the oxygen atoms in calcium carbonate. A multiple hydrogen bond structure is formed by two types of hydroxyl groups. Based on the adhesion phenomena and mechanism, the present work provides a design method for bonding materials, especially oxygen-containing minerals such as building materials. The current toxic and harmful traditional bonding materials are undoubtedly due to be replaced by the next generation of bonding materials with environmentally friendly advantages.

## Figures and Tables

**Figure 1 molecules-27-08179-f001:**
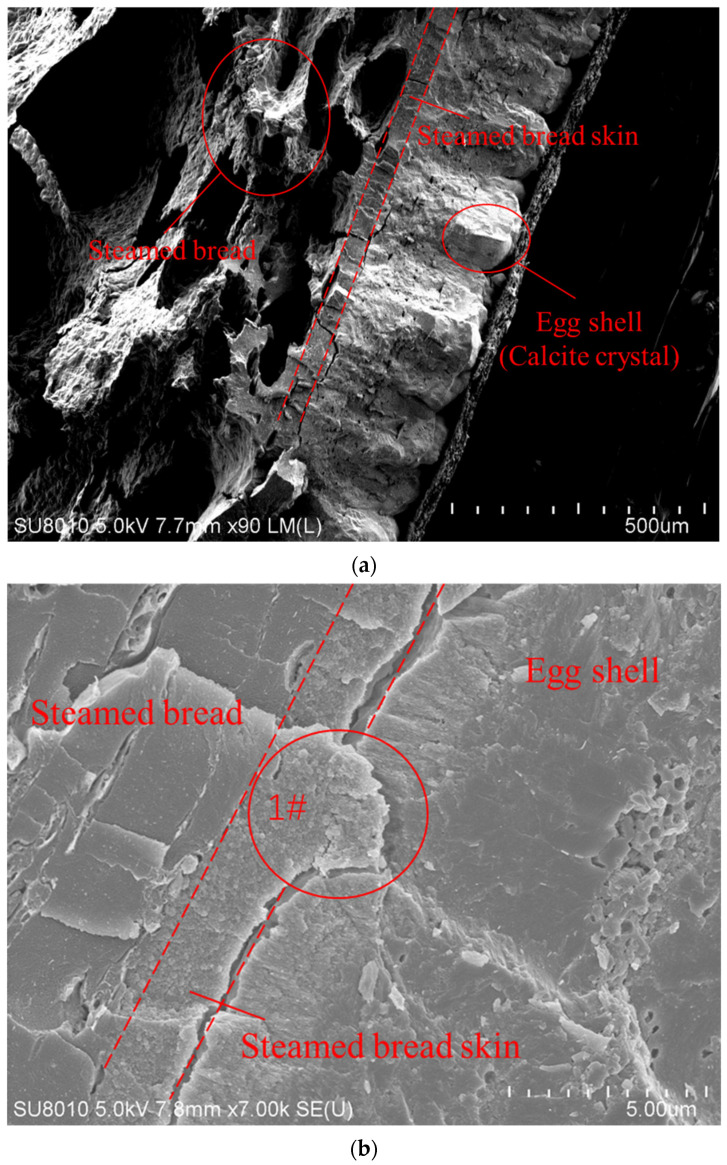
SEM image of the adhesion morphology of steamed bread and eggshell: (**a**) 90-fold overall topography of the adhesion interface. (**b**) 7000-fold topography (area 1# shows that the steamed bread skin expanded into the holes on the surface of the eggshell and increased the contact area). (**c**) 45,000-fold topography (area2# shows the morphology detail of the adhesion).

**Figure 2 molecules-27-08179-f002:**
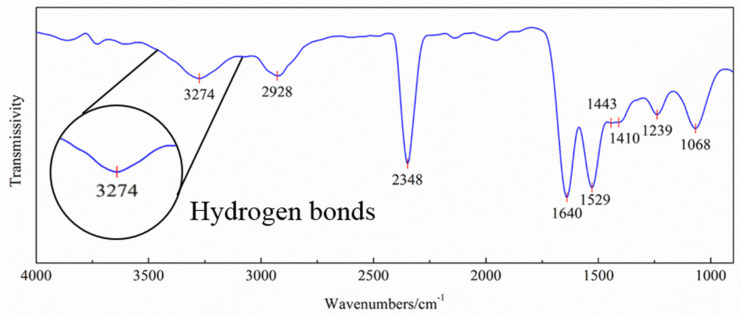
FTIR measurements of the interface.

**Figure 3 molecules-27-08179-f003:**
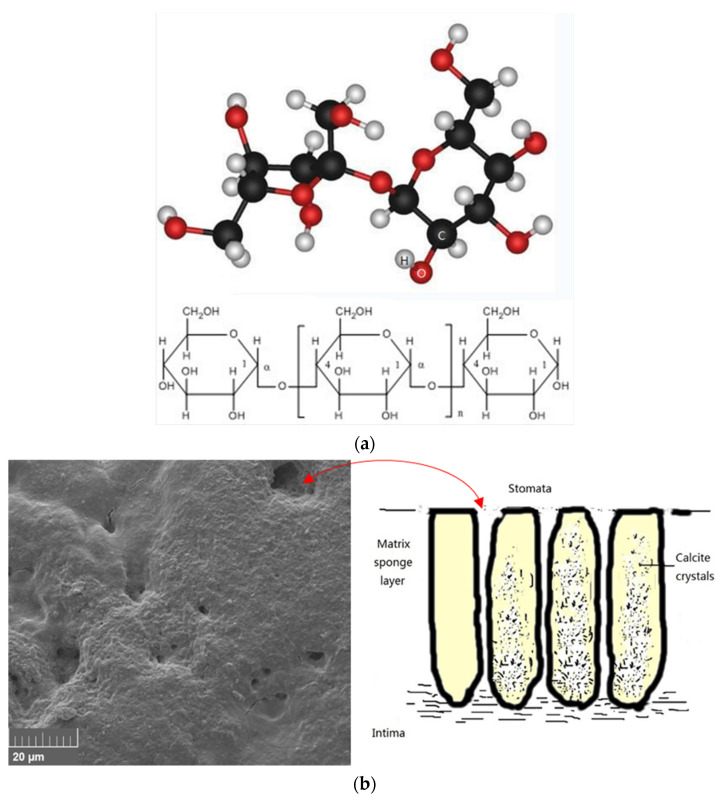
Microcosmic morphology, composition, and molecular model of starch and calcium carbonate. (**a**) Molecular model of starch. (**b**) Surface morphology and cross-section view of eggshell. (**c**) Results of EDS. (**d**) Molecular model of starch and calcium carbonate.

**Figure 4 molecules-27-08179-f004:**
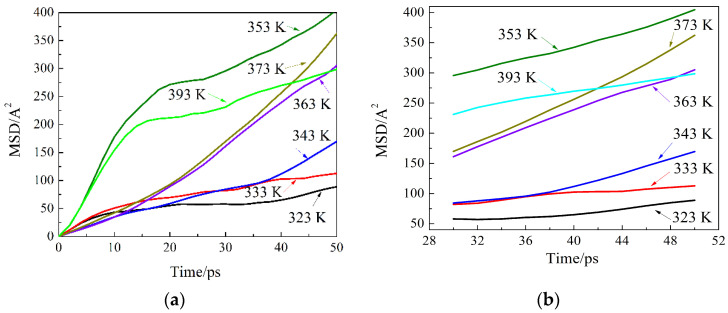
MSD curves of the simulation process:(**a**) MSD of the whole simulation progress;(**b**) MSD of the simulation after reaching equilibrium.

**Figure 5 molecules-27-08179-f005:**
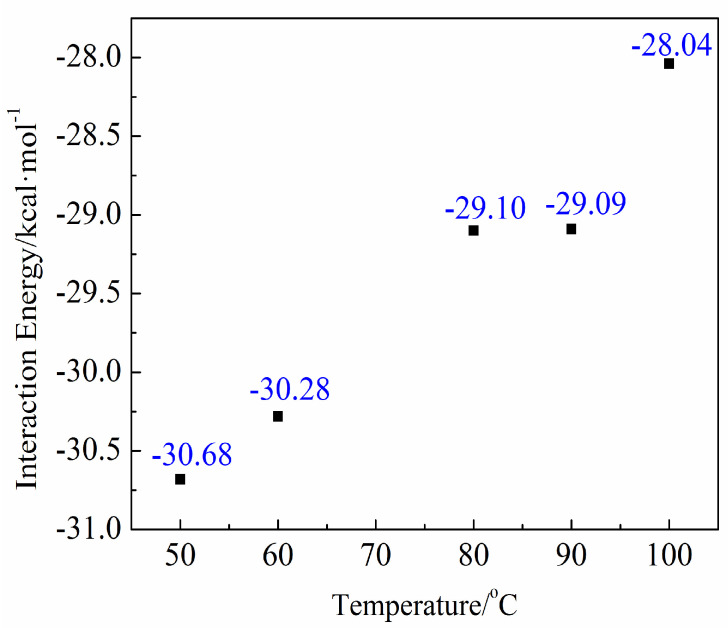
The interaction energy between the surface of the calcium carbonate and the starch molecules under different temperature conditions.

**Figure 6 molecules-27-08179-f006:**
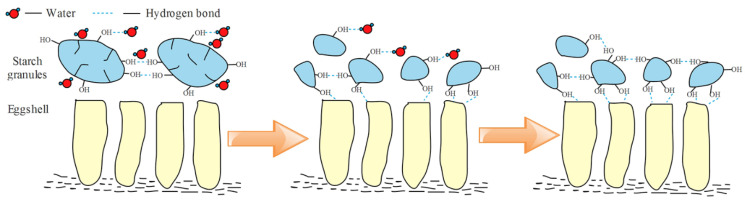
Schematic diagram of the bonding process.

**Figure 7 molecules-27-08179-f007:**
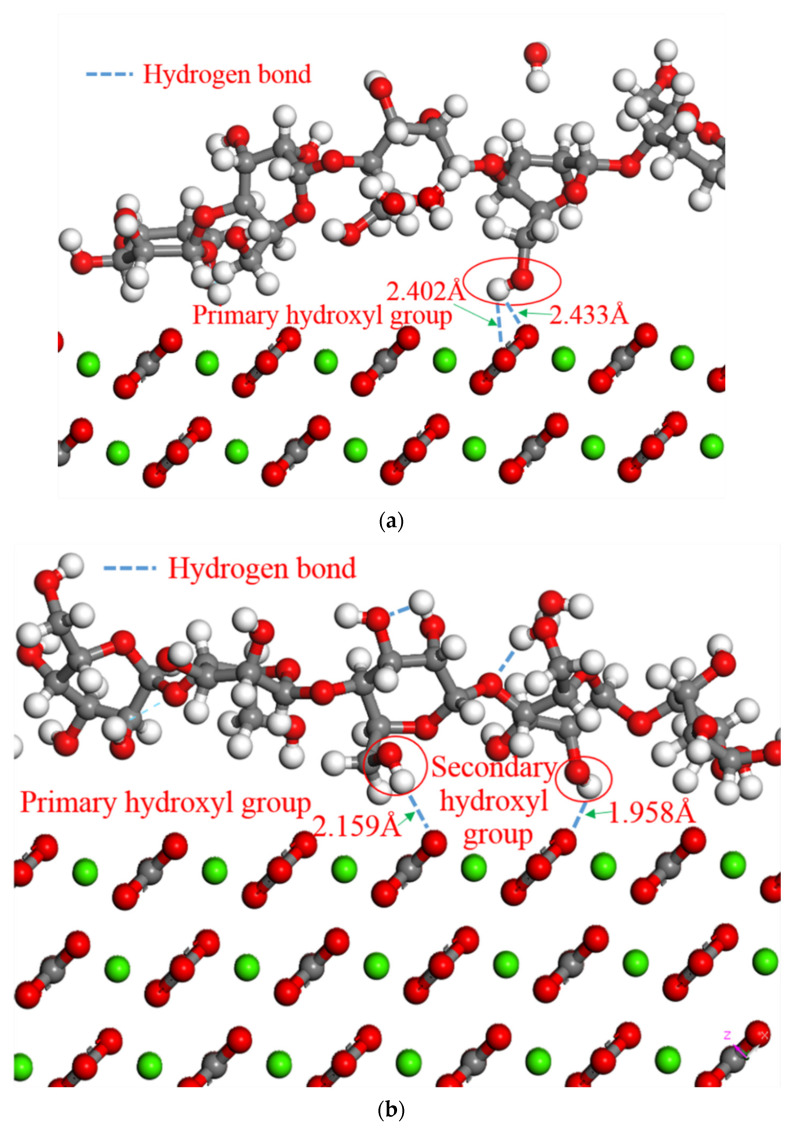
3D model and schematic representation of the simulation process: (**a**) 3D model of the hydrogen bonds formed by the primary hydroxyl groups and calcium carbonate before 18 ps; (**b**) 3D model of the hydrogen bonds formed by the primary hydroxyl groups and calcium carbonate after 18 ps; (**c**) schematic representation of the simulation process; (**d**) schematic diagram of primary hydroxyl groups and secondary hydroxyl groups.

**Figure 8 molecules-27-08179-f008:**
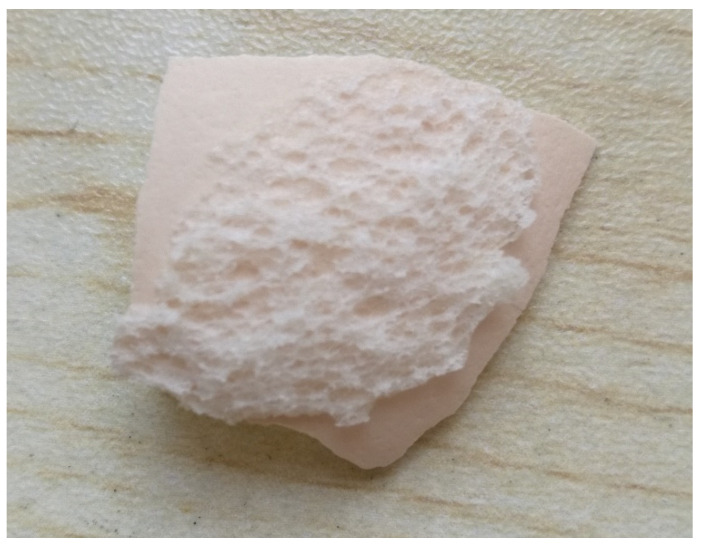
Image of the steamed bread and eggshell cooling immediately after steaming.

**Table 1 molecules-27-08179-t001:** Coefficient of diffusion.

Temperature/°C	Slope of MSD	Diffusion Coefficient/A^2^·ps^−1^
50	1.66	0.28
60	1.51	0.25
70	4.37	0.73
80	5.30	0.84
90	7.12	1.19
100	9.50	1.58
120	3.16	0.53

## Data Availability

All data, models, and code generated or used during the study appear in the submitted article.

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
