# Peer review of "Analysis of Adhesion at the Interface of Steamed Bread and Eggshell"

_molecules, 2022, doi:10.3390/molecules27238179_

Round 1

Reviewer 1 Report

In this work the authors studded the adhesion of the steamed bread and eggshell when the steamed bread and eggshells are contacted closely and cooled slightly in the ambient air.

The bread was heated at high temperature in steam from boiling water and put in contact, as closely as possible, with cooked egg in hot water, to ensure a good contacting between them. After that the sample was cooled.

At microscopic scale the adhesion is evaluated by scanning electron microscope.  The formation process and the mechanism of the strong-bond adhesion were also analyzed and discussed on the basis of the molecular dynamics simulation.

The strong adhesion of the steamed bread and eggshell system is attributed to the effect of the absorption, diffusion, surface chemistry and the formation of multiple hydrogen bonds between the starch contained by breath and the calcium carbonate crystals of the eggshell.

Different models of molecular packaging were proposed for simulation.

The simulation shows that the starch molecule diffuses in calcite crystal and forms multiple hydrogen bonds.

The subject of the work is interesting, but some improvement are necessarily.

The caption of figure 1 must be changed: The image of the steamed bread….instead The diagram of the steamed bread…

The caption of figure 2, SEM image of….. instead of SEM photograph….

How the starch and calcite were identified in the SEM images? All the elements of these images seem to have the same appearance.

The microstructure of the adhesion interface of the steamed bread and eggshell was measured using FTIR…The microstructure cannot be measured by FTIR.. This technique offers information about the vibration of chemical bonds that can be used to characterize the microstructure. The sentence must be reformulated. This analyze is not convincing, it is a simple spectrum of a sample that cannot demonstrate any shift of the vibration bands or any interaction between the calcite and the starch. To prove some modification on the microscopic scale, the FTIR analyze must be done on each component, first the bread and eggshell considered separately, before adhesion and after adhesion. Then the shift of the vibration bands, if these phenomena appear, must be discussed and correlated with the microstructure and the formation of chemical bonds.

In figure 4b the axis are not labeled. What this figure represents?

As I understood, the simulation allows the authors to calculate the diffusion coefficient. How can be correlate the values of this coefficient with the adhesion process of the starch to calcite?

Figure 4 contains two images, a spatial representation of the molecule and a flat image. What represents the second image?

How were calculated or measured the energies used in formula 6, Einteraction=Etotal-(Esurface+Epolymer) and represented in figure 6. What are the measurement units for the energy? How the values of energy can be correlated with the strength of the adhesion?

Usually, every simulation must be confirmed by comparison with some experimental data, in order to establish the validity of the model used for simulation. Did you done such kind of verification?

The strength of the adhesion must be investigate by rheological methods and then compared with the simulation.

What is the influence of the temperature on the adhesion strength?

English must be improved.

Author Response

Author's Reply to the Review Report (Reviewer 1)

Review Report Form

Comments and Suggestions for Authors

In this work the authors studied the adhesion of the steamed bread and eggshell when the steamed bread and eggshells are contacted closely and cooled slightly in the ambient air.

The bread was heated at high temperature in steam from boiling water and put in contact, as closely as possible, with cooked egg in hot water, to ensure a good contacting between them. After that the sample was cooled. At microscopic scale the adhesion is evaluated by scanning electron microscope. The formation process and the mechanism of the strong-bond adhesion were also analyzed and discussed on the basis of the molecular dynamics simulation. The strong adhesion of the steamed bread and eggshell system is attributed to the effect of the absorption, diffusion, surface chemistry and the formation of multiple hydrogen bonds between the starch contained by breath and the calcium carbonate crystals of the eggshell. Different models of molecular packaging were proposed for simulation. The simulation shows that the starch molecule diffuses in calcite crystal and forms multiple hydrogen bonds. The subject of the work is interesting, but some improvement is necessarily.

  1. The caption of Figure 1 must be changed: The image of the steamed bread….instead the diagram of the steamed bread.

Response: The authors appreciate the reviewers very much for providing the important suggestions about this manuscript. The authors replace the sentence with “The image of the steamed bread and egg shells cooling immediately after steaming”.

  1. The caption of figure 2, SEM image of….. instead of SEM photograph….

Response: The authors replace the sentence with “SEM image of the adhesion morphology of steamed bread and eggshell”.

  1. How the starch and calcite were identified in the SEM images? All the elements of these images seem to have the same appearance.

Response: The authors appreciate the reviewers. The observation process of the SEM images of the starch and calcite is listed as following. For qualitative observations of microstructure of the sample, after cooling of the starch and calcite, the morphology of sample of the steamed bread and egg shells was observed using scanning electron microscope (SEM, S-3000N, Japan) with an potential of 1 kV. The observations were made using a secondary electron detector. The photographs were taken using automatic image capture software. Firstly, the overall topography of the steamed bread and eggshell was measured by SEM to identify the adhesion interface with low resolution. Resolution is the most basic performance index of SEM. It is necessary that we should clarify some details of high resolution of SEM. Secondly, the location of the adhesion interface obtained from low resolution topography was observed under high resolution of 7000-fold to find that the steamed bread skin expands into the holes on the surface of the eggshell and increases the contact area). Finally, the adhesion of the steamed bread and eggshell was found under high resolution of 45000-fold to obtain details of the adhesion interface.

  1. The microstructure of the adhesion interface of the steamed bread and eggshell was measured using FTIR. The microstructure cannot be measured by FTIR. This technique offers information about the vibration of chemical bonds that can be used to characterize the microstructure. The sentence must be reformulated. This analyze is not convincing, it is a simple spectrum of a sample that cannot demonstrate any shift of the vibration bands or any interaction between the calcite and the starch. To prove some modification on the microscopic scale, the FTIR analyze must be done on each component, first the bread and eggshell considered separately, before adhesion and after adhesion. Then the shift of the vibration bands, if these phenomena appear, must be discussed and correlated with the microstructure and the formation of chemical bonds.

Response: The authors are grateful to the comments and suggestions. There are many types of vibrations in molecules, some of which can cause changes in molecular dipole distance. When the frequency of vibrations is the same as that of the infrared light, molecules can absorb the energy of the infrared light, forming the infrared absorption spectrum. Due to their different molecular structures, different compounds have different characteristic peaks of infrared absorption spectrum. There is no two compounds are completely consistent like human fingerprints. Therefore, FTIR is considered to be a very effective method in the analysis and identification of polymer materials. As reviewer said, this technique offers information about the vibration of chemical bonds that can be used to characterize the microstructure. In the structural identification, quantitative analysis and chemical kinetics research, its analysis can provide information about functional groups, the position and intensity of the infrared absorption peak reflecting the characteristics of molecular structure, thus FTIR can be used to identify the structure of the unknown or determine its chemical groups. Microscopic Fourier transform infrared spectrometer (FTIR) analysis is a kind of important modern analytical means and methods, and it has been widely used in the material evidence in judicial authentication materials (including organic and inorganic material evidence materials) samples of qualitative and quantitative analysis, not only it can accurately determine the material evidence materials of all kinds of chemical composition, but also adopt the method of comparative analysis quickly and efficiently. The microstructure of the adhesion interface of the steamed bread and eggshell was measured using FTIR to study the interesting and scientific phenomenon in our manuscript. FTIR analyze may be done on each component of the bread and eggshell in the many references as following. The authors have discussed and correlated with the microstructure and the formation of chemical bonds of the steamed bread and eggshell.

References:

  1. Boopalan M, Umapathy M, Jenyfer P (2012) A comparative study on the mechanical properties of jute and sisal fiber reinforced polymer composites. Silicon 4: 145-149.
  2. Zhang K, Zhang Z, Zhao M, Milosavljević V, Cullen P, Scally L, Sun D, Tiwari B (2021) Low-pressure plasma modification of the rheological properties of tapioca starch. Food Hydrocolloid 125:107380.
  3. Liu Q, Charlet G, Yelle S, Arul J (2002) Phase transition in potato starch–water system I. Starch gelatinization at high moisture level. Food Res Int 35: 397-407.
  4. Mujeeb M, Zafar M (2017) FTIR spectroscopic analysis on human hair. IntJInnovResSciEngTechnol6: 9327-9332.
  5. Ngarize S, Adams A, Howell N (2004) Studies on egg albumen and whey protein interactions by FT-Raman spectroscopy and rheology. Food Hydrocolloid18: 49-59.
  6. Huang C, Huang H, Qin P. In-situ immobilization of copper and cadmium in contaminated soil using acetic acid-eggshell modified diatomite [J]. Journal of Environmental Chemical Engineering, 2020, 8(4): 103931.
  7. Platon N, Georgescu A M, AruÅŸ V A, et al. Valorization of eggshells waste for bread production [J]. 2022.
  8. Zhang Y, Chen C, Chen Y, et al. Effect of rice protein on the water mobility, water migration and microstructure of rice starch during retrogradation[J]. Food Hydrocolloids, 2019, 91: 136-142.

9.Lu K, Zhu J, Bao X, et al. Effect of starch microstructure on microwave-assisted esterification[J]. International Journal of Biological Macromolecules, 2020, 164: 2550-2557.

  1. In Figure 4b the axis are not labeled. What this figure represents?

Response: In Figure 4b, the axis represents the acceleration voltage of the excited element. Figure 4b shows that the eggshell has a highly mineralized structure composed of the calcite crystals with many stomata distributing on the surface. According to the measurements of EDS, there are the elements of carbon, calcium and oxygen in the adhesion interface.

  1. As I understood, the simulation allows the authors to calculate the diffusion coefficient. How can be correlating the values of this coefficient with the adhesion process of the starch to calcite?

Response: According to the diffusion theory, it is difficult to obtain high adhesion strength only when the interfacial molecules are in directly contact with each other, and it is necessary to make the interfacial diffusion between the polymer molecular chains. The diffusion is that the network is formed by adhesion to the interface, which leads to the disappearance of the interface and the generation of the transition zone, and the two phases are connected through the winding of the diffused molecules or chain segments and their cohesion to form a strong adhesive joint. The diffusion coefficient depends on many factors. The diffusion is affected by the contact time and bonding temperature. When polymers are bonded to each other, higher temperature and longer time are, the stronger diffusion is. The molecular diffusion coefficient represents its diffusion capacity. According to Fick's law, the diffusion coefficient depends on the type of the diffused substance, the diffused medium and its temperature and pressure.

The molecular dynamics simulations are used to analyze the formation of hydrogen bond of the steamed bread and eggshell. The simulation provides the information of diffusion at the interface, allowing for molecular insight into the adhesion mechanism. In order to study the diffusion rate of the starch molecules on the surface of calcium carbonate under different temperatures, the mean square displacement of hydrogen atoms in starch molecules is analyzed to obtain the coefficient of diffusion and characterize the compatibility between starch and calcium carbonate. Therefore, the mean square displacement (MSD) curves of hydrogen atoms in starch molecules at these temperatures are obtained from the molecular dynamics simulations.

  1. Figure 4 contains two images, a spatial representation of the molecule and a flat image. What represents the second image?

Response: The second image in Figure 4 shows the surface morphology and cross section view of egg shells in details. The eggshell has a highly mineralized structure composed of the calcite crystals with many stomata distributing on the surface.

  1. How were calculated or measured the energies used in formula 6, Einteraction=Etotal-(Esurface+Epolymer) and represented in figure 6. What are the measurement units for the energy? How the values of energy can be correlated with the strength of the adhesion?

Response: Surface energy is an important parameter, which is required to break the chemical bonds between molecules when creating a material surface. It takes more heat to cut through surface atoms since surface atoms have higher heat than atoms inside a substance. The calculation method of surface energy requires the structural optimization of the slab model after cutting. The principle of surface energy calculation is that the energy required to cut the block into crystalline surfaces is equivalent to the energy required to form two new surfaces. In the calculation of surface energy, the upper and lower surfaces should be optimized at the same time. The calculation principle of interface energy is the total potential energy of the whole structure is calculated first, and then it is cut in the middle to form two interfaces. After the two parts are completely separated, the new potential energy is calculated and put into the formula to get the interface energy. The interaction energy of the steamed bread and egg shell is calculated to characterize the stability of the bonding interface under different temperatures. The intermolecular interaction energy is calculated as the total energy of each system under a stable configuration. The action of bonding occurs between the interfaces in contact with each other. The adhesive is fully wet to the surface of the adhesive. The good wetting and bonding are necessary to achieve the formation of good adhesion force. Surface energy is an important factor affecting the adhesion strength. Reducing the surface energy or increasing the surface energy of the adhesive can enhance the wettability on the adhesive, thus improving the adhesion strength.

  1. Usually, every simulation must be confirmed by comparison with some experimental data, in order to establish the validity of the model used for simulation. Did you done such kind of verification?

Response: The simulations are used to analyze the formation of hydrogen bond of the steamed bread and eggshell after the experiment. The simulation in our manuscript provides the information of diffusion at the interface, allowing for molecular insight into the adhesion mechanism.

  1. The strength of the adhesion must be investigate by rheological methods and then compared with the simulation.

Response: As Reviewers said, the strength of the adhesion may be investigate by rheological methods and then compared with the simulation, which is helpful to understand the adhesive process of the steamed bread and the eggshell. However, the authors try to address the adhesive mechanism of the steamed bread and the eggshell after the authors observed this interesting phenomena.

  1. What is the influence of the temperature on the adhesion strength?

Response: Many environmental conditions affect the performance of adhesives. One of the most important environmental factors is temperature. A small change in temperature can have a significant impact on the rate of adhesion of the adhesive. Temperature affects the bonding strength of adhesives. Temperature and humidity affect the drying speed of glue. The higher the temperature, the faster the glue dry.

  1. English must be improved.

Response: The authors are heartily grateful to the Editor and Reviewers’ comments and suggestions. The authors have already tried our best to refine the language and improve the whole manuscript carefully and make reading easier according to the reviewers’ comments in the revised manuscript. 

Reviewer 2 Report

The idea presented in the manuscript is interesting, while some important points have been omitted, which is why I recommend the article for rejection. All comments are available below.

1) The first question that arises after reading this manuscript is:

Why did the authors choose bread made from wheat flour as research material to investigate the adhesive behavior of starch? It is well known that, in addition to starch and other carbohydrates, flour also contains ingredients such as protein (including gluten proteins), fat and fiber. The above components will also have an impact on the phenomenon under investigation, and unfortunately their effect has not been taken into account.

2) Furthermore, the information on the material used is only rudimentary because:

Line 82-83 “The steamed bread samples, which were bought from the local store… ”

Incomplete information about the material means that the experiment cannot be reproduced by other researchers. The reproducibility of an experiment is a basic requirement of the Aims of Molecules journal: “Full experimental details must be provided so that the results can be reproduced.

3) The authors assume that:

Line 98-99: “It is well known that the steamed bread is made by the wheat flour which is mixed by water and backing soda. ”

According to the reviewer, this is not the case, as such baked goods are also made with yeast. This is another point that has been completely neglected.

4) Chapter 2 Experimental Materials contains information that should not be included:

a) Line 73 – 76: There are many natural adhesive systems such as the gecko footpads (van der Waals  forces)  or  the  secretions  of  mussels  and  tubeworms  (hydrogen  bonds)  and  various biological  processes  including  cell  adhesion  to  extracellular  biomolecules  or  bacterial  adhesion [14].

If the above information is relevant, it should be published in the Introduction.

b) Line 76-77: „In this study, it reports a research about the potential of starch adhesives in the role of nature as a source of inspiration.”

The above sentence does not identify the material used, but indicates a source of inspiration that should also be included in the introduction.

(c) Subsections 2. 1. Analysis of the structure and 2. 2. Molecular Dynamic Simulations should be listed in Chapter/Subsection Methods/Experimental.

5) The description of preparing the sample of bread and eggs for examination is laconic. It is not known which part of the bread was selected for sample preparation (shell, pulp, top or bottom of the loaf). It’s important. There is also no information on the number of repetitions, which indicates that there was only one.

6) Line 136: “Put the bread and eggs together and steam them. What does this sentence refer to? I have a feeling it is in this place by chance.

7) Line 138: “The Cross-Section is gold-sprayed This information should also be included in the description of the preparation of the sample for examination. Unfortunately, this description is missing from the manuscript.

8) SEM photos of the egg shell itself are missing.

9) Line 169-171: “The wavelength is 1529 cm-1 has been observed, which can be adapted to the stars in the region of proteins [20]. ” The statement “the starch peak region of proteins” is incorrect as the proteins are not derived from starch but are contained only in the wheat flour used in the production of the bread used. Furthermore, a reference is not appropriate, since there are many publications in which bakery products are used as research material.

10) Line 189:  ”…the steamed bread and eggshell was closed.” It should be “close. ”

11) Line 189-191: ”Starch is a hydrophilic natural polymer and can easily absorb up to more 50% water molecules on a dry weight basis when steamed in water vapor.” Yes, but In this case, it is not pure starch, but flour. It should also be mentioned that steamed bread was used, so that the starch present in the flour became sticky during the manufacturing process. During the experiment, it is treated a second time with steam.

12) Line 191: “As expected, the starch, especially in skin, was fully soggy with water vapor during steaming.” On what basis was this finding made?

13) Line 201-212: The manuscript does not specify to which data equation 1 was applied. There is no description of the methodology for collecting the data needed to use this equation.

14) Line 213-219: This paragraph mentions the production process of Steamed Bred and the ingredients from which it is made. The description does not correspond to the information given in the manuscript (line 98-99).

15) Line 239: Why was a publication with pea starch chosen as a reference? [28]

16) Rys 4 c) description of axes and units is missing. The lack of potassium and phosphorus in the composition is also surprising.

17) Line 294: The recording of the equation needs to be improved.

18) Figure 7 shows the schematic diagram of the bonding process. Description (Line 317-318) refers to the preservation of native starch (starch slurry). After all, the authors did not use native starch for the adhesion test, but only an already starch after pasting.

19) Finally lines 254-257: “Initially, a distance of 4.0 Å  is used maintained between the starch molecule and the calcite crystals to avoid overlap. A vacuum space of  40 Å  is added on top of the starch molecule to restrict its interaction only to the uppermost atom layer of cellulose and also enhance computational efficiency.” This section was copied from an unquoted publication (section Prepare Molecular Model of System) and contains the word “cellulose” as an error, because Authors used starch.

Mir A. A. R. Quddus, Orlando J. Rojas, and Melissa A. Pasquinelli

Molecular Dynamics Simulations of the Adhesion of a Thin Annealed Film of Oleic Acid onto Crystalline Cellulose, Biomacromolecules 2014, 15. https://doi.org/10.1021/bm500088c

Author Response

Author's Reply to the Review Report (Reviewer 2)

Review Report Form

Comments and Suggestions for Authors:

The idea presented in the manuscript is interesting, while some important points have been omitted, which is why I recommend the article for rejection. All comments are available below.

1) The first question that arises after reading this manuscript is:

Why did the authors choose bread made from wheat flour as research material to investigate the adhesive behavior of starch? It is well known that, in addition to starch and other carbohydrates, flour also contains ingredients such as protein (including gluten proteins), fat and fiber. The above components will also have an impact on the phenomenon under investigation, and unfortunately their effect has not been taken into account.

Response: Thank you very much for your’ comments and suggestions. In the present paper, an adhesion system of the steamed bread and eggshell is found when the steamed bread and eggshells are contacted closely and cooled slightly in the ambient air. The authors have discovered this phenomenon by accident. And then the adhesion phenenoma and mechanism of the interfaces at the steamed bread and eggshell was investigated and discussed. The authors did not choose bread made from wheat flour as research material to investigate the adhesive behavior of starch specially. The components of gluten proteins maybe have an impact on the phenomenon under investigation, however, the effect of ingredients may be limited.

2) Furthermore, the information on the material used is only rudimentary because:

Line 82-83 “The steamed bread samples, which were bought from the local store… ”

Response: The steamed bread samples were bought from the local store. The steamed bread mainly contains carbohydrates, crude fiber, protein and manganese and other substances.

Incomplete information about the material means that the experiment cannot be reproduced by other researchers. The reproducibility of an experiment is a basic requirement of the Aims of Molecules journal: “Full experimental details must be provided so that the results can be reproduced.”

Response: The authors appreciate the reviewers very much for providing the suggestions. The authors reproduced these experiments many times when I cooked the bread and eggshell every morning, and then cooled the steamed bread and eggshell closely contact, and we found these interesting phenenoma every time, as you know, we need have a good breakfast, why not you try, you also can achieve this interesting phenenoma.

3) The authors assume that:

Line 98-99: “It is well known that the steamed bread is made by the wheat flour which is mixed by water and backing soda. ” According to the reviewer, this is not the case, as such baked goods are also made with yeast. This is another point that has been completely neglected.

Response: As Reviewer said, it is not the same case. The author have cooked the bread and eggshell in the heating water, and then cooled the steamed bread and eggshell closely contact in the ambient air, and we found this interesting phenenoma every time. It is different from the bake goods, we do not cook these baked goods in the heating water.

4) Chapter 2 Experimental Materials contains information that should not be included:

  1. a) Line 73-76: There are many natural adhesive systems such as the gecko footpads (van der Waals forces) or  the  secretions  of  mussels  and  tubeworms  (hydrogen  bonds)  and  various biological  processes  including  cell  adhesion  to  extracellular  biomolecules  or  bacterial  adhesion [14]. If the above information is relevant, it should be published in the Introduction.

Response: The authors think that the above information is relevant with our adhesion system, it is good to be published in the introduction.

  1. b) Line 76-77: In this study, it reports a research about the potential of starch adhesives in the role of nature as a source of inspiration.” The above sentence does not identify the material used, but indicates a source of inspiration that should also be included in the introduction.

Response: The materials are starch from the steamed bread and calcium carbonate from eggshell, which is conventional materials, including in the manuscript.

(c) Subsections 2. 1. Analysis of the structure and 2. 2. Molecular Dynamic Simulations should be listed in Chapter/Subsection Methods/Experimental.

Response: The authors changed with subsection 2.1 and 2.2 to be listed in Methods and experimental.

5) The description of preparing the sample of bread and eggs for examination is laconic. It is not known which part of the bread was selected for sample preparation (shell, pulp, top or bottom of the loaf). It’s important. There is also no information on the number of repetitions, which indicates that there was only one.

Response: Section 2.1 shows the description of preparing the sample of bread and eggs for experiments. The shell of the bread was selected for sample preparation after cooling. The authors reproduced these experiments many times when I cooked the bread and eggshell every morning, and then cooled the steamed bread and eggshell closely contact, and we found these interesting phenenoma every time.

6) Line 136: “Put the bread and eggs together and steam them. ” What does this sentence refer to? I have a feeling it is in this place by chance.

Response: The authors place the eggs in the water and the bread on the steamer one by one, leaving half a space between each one, and heating water to boiling for 15-20 minutes. After heating, the authors take out the eggs and bread, and then place them closely contact each other to cool in the ambient air. It is not in this place by chance.

Figure 1 Image of the steamed bread

7) Line 138: “The Cross-Section is gold-sprayed… ” This information should also be included in the description of the preparation of the sample for examination. Unfortunately, this description is missing from the manuscript.

Response: The authors moves this information to Section 2 as the description of the preparation of the sample for examination in the manuscript.

8) SEM photos of the egg shell itself are missing.

Response: The observation process of the SEM images of the starch and calcite is listed. The section of the interface of sample of the steamed bread and egg shells was observed using SEM. The outermost layer of the eggshell is dense with certain strength, as shown in Figure 2. There are holes about 30 microns in diameter at the surface of the eggshell, which are the exchanged channels of the gas between the inside and outside of the eggshell.

Figure 2 SEM images of eggshell: (a) low magnification (b) high magnification

9) Line 169-171: “The wavelength is 1529 cm-1 has been observed, which can be adapted to the stars in the region of proteins [20]. ” The statement “the starch peak region of proteins” is incorrect as the proteins are not derived from starch but are contained only in the wheat flour used in the production of the bread used. Furthermore, a reference is not appropriate, since there are many publications in which bakery products are used as research material.

Response: The authors appreciate the reviewers for providing the important suggestions. The authors have modified the sentence“The wavelength is 1529 cm-1 has been observed, which can be adapted to the wheat flour peak region of proteins [20]. ” The authors have modified this reference according to Reviewer’s suggestions.

Kamble D, Rani S. Bioactive components, in vitro digestibility, microstructure and application of soybean residue (okara): A review[J]. Legume Science, 2020, 2(1): e32.

10) Line 189: “…the steamed bread and eggshell was closed.” It should be “close. ”

Response: The authors have modified this sentence according to Reviewer’s suggestions.

11) Line 189-191: “Starch is a hydrophilic natural polymer and can easily absorb up to more 50% water molecules on a dry weight basis when steamed in water vapor.” Yes, but In this case, it is not pure starch, but flour. It should also be mentioned that steamed bread was used, so that the starch present in the flour became sticky during the manufacturing process. During the experiment, it is treated a second time with steam.

Response: The authors appreciate the reviewers for suggestions. The authors have modified this sentence according to Reviewer’s suggestions. The authors have modified this sentence. Starch is a hydrophilic natural polymer. However, flour in the bread can easily absorb up to more 50% water molecules on a dry weight basis when steamed in water vapor. The steamed bread was used in the manuscript, so that the starch present in the flour became sticky during the manufacturing process. During the experiment, it is treated a second time with steam.

12) Line 191: “As expected, the starch, especially in skin, was fully soggy with water vapor during steaming.” On what basis was this finding made?

Response: The authors have modified this sentence. During steaming bread, we can find that the surface of the starch was fully soggy with water vapor, as show in Figure 3.

Figure 3 Images of the steamed bread with soggy

13) Line 201-212: The manuscript does not specify to which data equation 1 was applied. There is no description of the methodology for collecting the data needed to use this equation.

Response: Fick's second law, which predicts the change in concentration with time due to diffusion, is a parabolic partial differential equation. The diffusion coefficient D is an important physical quantity describing the diffusion velocity. The larger the value of D is, the faster the diffusion is. Fick's second diffusion equation describes the change of the concentration of matter at each point in the medium due to diffusion under the condition of unstable diffusion. The law of material concentration variation with time and location can be obtained by solving Fick's second diffusion equation according to the initial conditions and boundary conditions. The data of time and concentration was applied to equation 1.

14) Line 213-219: This paragraph mentions the production process of Steamed Bred and the ingredients from which it is made. The description does not correspond to the information given in the manuscript (line 98-99).

Response: Line 213-219, this paragraph shows the production process of the steamed bread in the first time. The steamed bread was used in the manuscript, it is treated a second time with steam during the experiment line 98-99.

15) Line 239: Why was a publication with pea starch chosen as a reference? [28]

Response: The authors make mistake to choose this reference and replace this reference with new reference, Xu K, Chi C, She Z, et al. Understanding how starch constituent in frozen dough following freezing-thawing treatment affected quality of steamed bread[J]. Food Chemistry, 2022, 366: 130614.

16) Rys 4 c) description of axes and units is missing. The lack of potassium and phosphorus in the composition is also surprising.

Response: In Figure 4c, the axis represents the acceleration voltage of the excited element (eV). The potassium and phosphorus in the composite were not found in EDS because there is not potassium and phosphorus in the observed location. According to the measurements of EDS, there are the elements of carbon, calcium and oxygen in the adhesion interface.

17) Line 294: The recording of the equation needs to be improved.

Response: The equation is correct in the word manuscript, as shown in the following.

                         (4)

18) Figure 7 shows the schematic diagram of the bonding process. Description (Line 317-318) refers to the preservation of native starch (starch slurry). After all, the authors did not use native starch for the adhesion test, but only an already starch after pasting.

Response: Figure 7 just show schematic diagram of the bonding process. The authors modified this sentence. Tthe starch slurry in the steamed bread is heated and the water molecules are entered into the amorphous regions of the starch granules in the steamed bread, and then the hydrogen bonds are broken among the starch molecules to eliminate the association in the starch molecular chain.

19) Finally lines 254-257: “Initially, a distance of 4.0 Å is used maintained between the starch molecule and the calcite crystals to avoid overlap. A vacuum space of 40 Å is added on top of the starch molecule to restrict its interaction only to the uppermost atom layer of cellulose and also enhance computational efficiency.” This section was copied from an unquoted publication (section Prepare Molecular Model of System) and contains the word “cellulose” as an error, because Authors used starch.

Response: The authors describe the setting process of simulation. Initially, the distance of the starch molecule and the calcite crystals is 4.0 Å. A vacuum space on top of the starch molecule is 40 Å to restrict its interaction with the uppermost atom layer of starch and also enhance computational efficiency. The authors cited this reference. Quddus M A A R, Rojas O J, Pasquinelli M A. Molecular dynamics simulations of the adhesion of a thin annealed film of oleic acid onto crystalline cellulose[J]. Biomacromolecules, 2014, 15(4): 1476-1483.

Round 2

Reviewer 1 Report

I agree the supplementary information given by the authors. 

Reviewer 2 Report

Dear Authors,

Thank you for all changes and answers.

Please restructure the sentence as suggested below:

11) Line 189-191: “Starch is a hydrophilic natural polymer and can easily absorb up to more 50% water molecules on a dry weight basis when steamed in water vapor.” Yes, but In this case, it is not pure starch, but flour. It should also be mentioned that steamed bread was used, so that the starch present in the flour became sticky during the manufacturing process. During the experiment, it is treated a second time with steam.

Starch is a hydrophilic natural polymer and can easily absorb up to more 50% water molecules on a dry weight basis when steamed in water vapor.  The gelatinized starch, which forms the structure of bread (among others with gluten proteins), especially in skin, was fully soggy with water vapor during steaming.

The eqation 4 is still bad quality. Please, remember about this in proof corections.